# Mapping spatiotemporal distribution of forest carbon density in Xizang, China

**Li Cheng** [ID][1,2,3], **Zi ling Yang**[1,2,3‡], **Yang yang Xia**[1,4‡], **Wen wen Guo**[1,2‡], **Rui qiang Ren**[5‡], **Jiang ping Fang**[1,2*]

**1** School of Ecology and Environment, Xizang University, Lhasa, Xizang, China, **2** Key Laboratory of Ecological Security in the Yarlung Zangbo River Basin, Xizang Autonomous Region, School of Ecology and Environment, Xizang University, Lhasa, Xizang, China, **3** Key Laboratory of Biodiversity and Environment on the Qinghai-Tibetan Plateau, Ministry of Education, School of Ecology and Environment, Xizang University, Lhasa, Xizang, China, **4** Plateau Major Infrastructure Smart Construction and Resilience Safety Technology Innovation Center, Lhasa, Xizang, China, **5** Lhasa Municipal People's Government, Lhasa, Xizang, China

‡ ZlY, YyX, WwG and RqR also contributed equally to this work.
* xzfjp@sina.com

**Data availability statement:** Data Availability Statement: The data underlying the results presented in the study are available from Google

## Abstract

Climate warming is a major global challenge, and forests, essential carbon sinks, are critical in mitigating its effects. Forest carbon density is a key parameter in assessing the carbon sinks. Traditional estimating methods of forest carbon density are time-consuming, labor-intensive, and difficult to apply on a large scale. Combining multispectral data with machine learning offers a promising solution, but accurately estimating forest carbon density remains challenging due to the band limitations of multi-spectral data. This study proposes a novel approach to address this limitation gap. We utilized Landsat 8 data and 919 samples from Xizang, China, simultaneously constructed geographic (GEO) and environmental factors (GEF) for estimating forest carbon density for the first time, and adopted three models to evaluate the effectiveness. The results indicate that the extreme gradient boosting (XGB) model is significantly better, the average $R^2$ exceeds 0.77, especially in Rikaze exceeds 0.96. The total relative importance of GEF in the modelling exceeded 60%, Geo was the most critical variable, followed by CI. This study successfully used multi-spectral data to quantify the spatiotemporal distribution of forest carbon density and demonstrated that GEO and GEF are indispensable, which is expected to provide new perspectives and technical support for global carbon sink monitoring.

## Introduction

Climate change has become one of the global community's most pressing issues [1–4]. As the most significant carbon reservoir in terrestrial ecosystems, forests are irreplaceable in mitigating climate change [5,6]. Trees are the main component of forest ecosystems [7]. They

Earth Engine (https://developers.google.cn/earth-engine), which provides access to Landsat 8 satellite data. The data is available under the CC BY 4.0 license. For more information, please visit the Google Earth Engine website: https://developers.google.cn/earth-engine.

**Funding:** Authors: LC, ZLY, YYX, WWG, RQC, JPF. Project title and grant number: Spatial Patterns of Carbon Sinks in Tibetan Forest Ecosystems and the Factors Affecting Them (XZ202401ZY0090); Xizang University Graduate High-level Talent Cultivation Research Fund Project (2022-GSP-B011). Funding unit: Xizang Science and Technology Agency; Xizang University.

provide a variety of ecological services and can sequester and store carbon, thereby significantly affecting the carbon balance of the ecosystem [8,9]. From 2015 to 2020, forest cover decreased by 10 million hectares per year due to urbanization, human resettlement, and agricultural encroachment [10,11]. Xizang is considered one of China's most important forest reserves [12,13], and its importance to the carbon cycle is self-evident. Therefore, monitoring the forest carbon density of the Tibetan Plateau is crucial to improving the global forest carbon sink capacity and ensuring a safe and livable environment for human beings in the future.

Traditional methods for estimating forest carbon stocks include directly or indirectly measuring forest biomass and then calculating based on the carbon content in the biomass [14]. The main types of forest carbon stock estimation methods in recent years can be summarized into three categories [15,16]: Inventory-based estimates, satellite-based estimates, and process-based estimates [17]. Inventory-based methods use regional forest inventory data to estimate forest carbon stocks, including forest type, stand age, stand density, stand volume, average tree height, and diameter at breast height [18,19]. Satellite-based methods estimate forest carbon stocks by building relationships between band combinations and stand volumes, using optical remote sensing data and synthetic aperture radar satellite data (SAR) [20–22]. There are two main types of process-based approaches: geostatistical modeling and mechanistic modeling. Geostatistical modeling combines forest inventory data with environmental factors such as topography, elevation, slope, and aspect to construct statistical models to estimate forest biomass at a regional scale [23,24]. Mechanistic modeling quantitatively estimates forest carbon storage through mass-energy exchange or biogeochemical processes [25]. Google Earth Engine (GEE) [26] is an open-access remote sensing tool and cloud database that contains trillions of satellite images of different resolutions captured by multispectral satellites. Therefore, using multispectral satellite data to estimate forest carbon density has the advantages of open access, high temporal resolution, and comprehensive coverage. However, due to limited spectral information, accurately estimating forest carbon density using multispectral data still faces challenges [27].

Factors such as forest age, climate, and soil have significant impacts on forest carbon density [28]. In addition to these factors, topography (such as slope and aspect) and soil properties are also key factors affecting the distribution of forest carbon stocks [29]. Topography and soil properties affect canopy density and forest age indirectly or directly by changing vegetation growth factors, which are the most critical drivers of vegetation carbon storage [30]. Therefore, topography and soil properties may be crucial in revealing carbon storage at the regional scale. In addition, differences in altitude directly affect forest carbon density [31], which is mainly due to the increase in organic matter content and decreases in soil bulk density caused by suitable temperature [32–34], rich biodiversity [35,36], and careful management measures such as fertilization and soil loosening [37,38], which promote the increase of organic matter content and the reduction of soil bulk density, making trees grow more luxuriantly and improving carbon density. Therefore, the growth differences of different species along altitude and latitudinal gradients may provide valuable information for estimating forest carbon density [39]. However, the combination of geographic and environmental factors (GEF, geographic information and environment auxiliary variables) in estimating forest carbon density shows a gap. Therefore, the study of forest carbon density estimation using GEF is critical and urgent.

To fill this gap, this study combines Landsat 8 satellite images with machine learning algorithms to estimate forest carbon density. To this end, we used 919 samples from Xizang, China. Considering the adequate characterization of environmental factors by band information, spectral indices, and environmental auxiliary variables in Landsat 8 data, this study

will use these variables and the geographic information of the sample points to characterize GEFs. The research results are expected to provide valuable reference and technical support for global forest carbon density monitoring and enhance the practicality of multispectral data in environmental science applications.

## Materials and methods

### Study area

As shown in Fig 1, Xizang is located in the southwest of China, with geographical coordinates of 26°50' to 36°53' north latitude and 78°25' to 99°06' east longitude, and an average altitude of more than 4,000 meters. Affected by topography, landforms, and atmospheric circulation, the climate in the region is characterized by severe cold and dryness in the northwest and warm and humid weather in the southeast. The average annual temperature is about 8°C, with the lowest monthly average temperature dropping to -16°C and the highest monthly average temperature exceeding 16°C. The annual precipitation is between 74.8 and 901.5 mm, with precipitation mainly concentrated from June to September, accounting for 80% to 90% of the annual precipitation. The permanent population is 3.65 million. The forest area is 14.9099 million hectares, and the forest stock is 2.283 billion cubic meters, ranking fifth and first in China respectively.

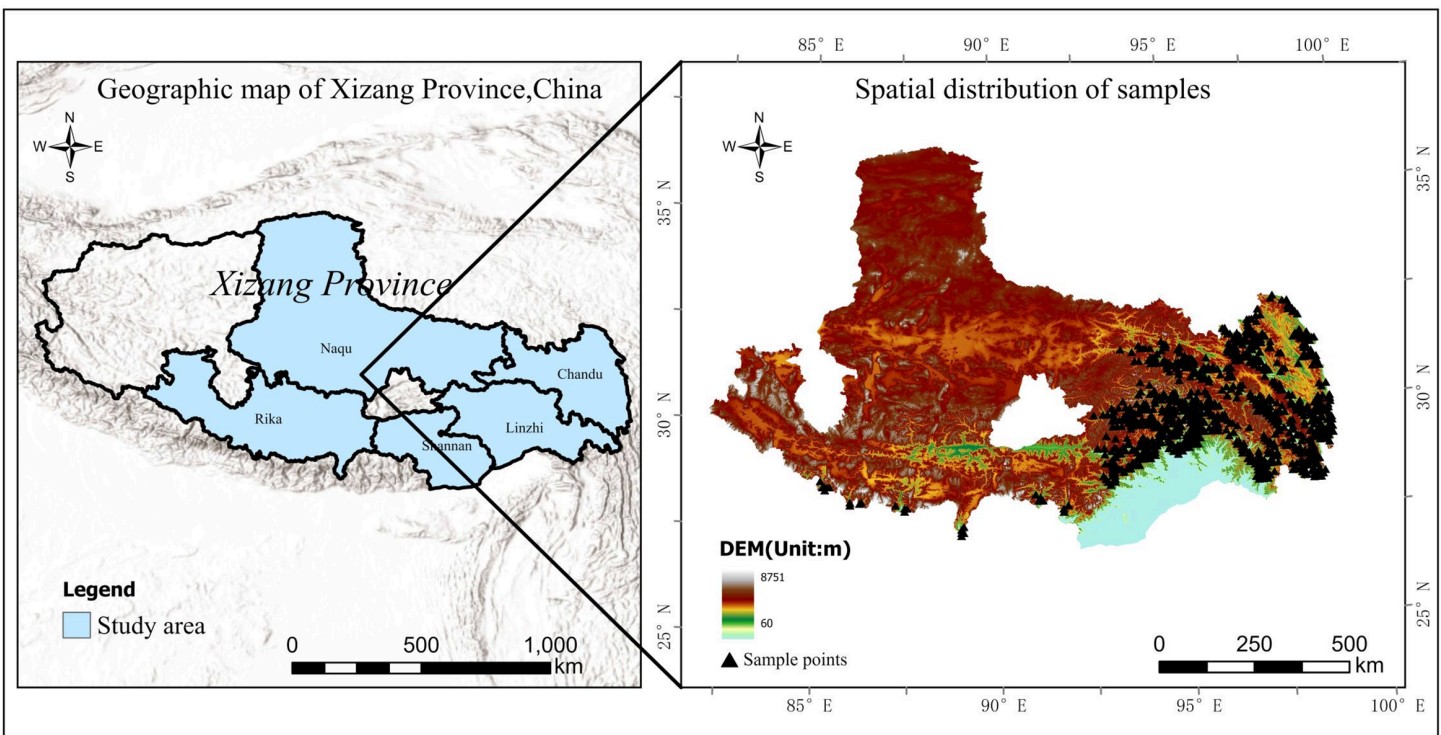

**Fig 1. Geographical location of samples in Xizang, China.** The figure shows two maps: the base map on the left shows the Xizang provincial boundary and is sourced from Natural Earth (public domain; https://www.naturalearthdata.com); the map on the right shows a digital elevation model (DEM) hillshade, sourced from USGS SRTMGL1 V003 (public domain; https://lpdaac.usgs.gov/products/srtmgl1v003/). Sampling areas are marked in blue, and black dots indicate the sampling points.

## Sample acquisition and statistical properties

From May to the end of June 2016, the system collected a total of 919 samples, with a total sampling area of 836,640 square kilometers, and recorded the GPS coordinate information of each sample. The sampling standard is that the measurement is carried out at a height of 1.3 meters from the upslope root neck of the tree trunk. The measurer should stand in the upslope position of the sample tree, ensure that the circumference is perpendicular to the central axis of the trunk, and tighten it before reading the data. All trees' diameter at breast height is measured using a steel circumference ruler, and the reading is accurate to 0.1 cm. For samples with a breast diameter of more than 60 cm, if it is impossible to measure with a steel circumference ruler, use a tape measure to measure the circumference at breast height and convert it into diameter. Before measuring the ruler, the vines, mosses, and other attachments on the trunk should be removed, and the fallen bark on the coniferous trees should also be removed before measurement. The forest carbon density was then calculated in the laboratory using the data obtained from the sampling. Due to the wide distribution of samples, we divided the study area into five regions (Shannan (28), Rikaze (19), Naqu (21), Linzhi (465), and Changdu (386)) according to the municipal administrative regions.

## Landsat 8 multi-spectral data

Due to a malfunction of the scan line corrector of Landsat 7 ETM+, all data after May 31, 2003, are defective. Therefore, this study uses Landsat 8 data (Collection 2, Tier 1, and Level 2) obtained through Google Earth Engine [26], which has undergone geometric correction and atmospheric correction. It covers coastal B1, visible light (blue band (B2), green band (B3), and red band (B4)), near-infrared (B5), and shortwave infrared (B6, B7). The resolution of all bands is 30 m. Due to each region's vast area, the data date span is relatively large to obtain complete images. However, we controlled each year to be consistent, which makes the years comparable. The specific dates of the images are shown in Table 1.

## Estimating models

The nonlinear models include The Extreme Gradient Boosting (XGB) [40] and the Random Forest Regression (RFR) [41] models. The XGB is an ensemble learning technique founded on the principles of gradient-boosting machines. Hyperparameters include `max-depth`, `min-child-weight`, `gamma`, `subsample`, `colsample-bytree`, `eta`, and `lambda`. Moreover, the RFR, a well-established ensemble learning approach, enhances prediction models' accuracy and robustness by incorporating multiple decision trees. Hyperparameters include `trees`, `leaf`, and `fboot`.The linear models include the Partial Least Squares Regression (PLSR) [42]; it serves effectively in addressing the linear relationship between the predictor variable $X$ and the response variable $Y$.

**Table 1. Dates of Landsat 8 images.**

| Item | Image dates | | | | |
| --- | --- | --- | --- | --- | --- |
| | Linzhi City | Changdu City | Rika City | Shannan City | Naqu City |
| Landsat 8 | 2016.8.3–2016.11.23 | 2016.5.24–2016.11.23 | 2016.5.4–2016.10.20 | 2016.8.3–2016.11.21 | 2016.5.4–2016.11.29 |
| | 2019.6.2–2019.11.27 | 2019.6.2–2019.12.13 | 2019.5.6–2019.11.24 | 2019.6.2–2019.11.23 | 2019.5.2–2019.11.23 |
| | 2022.7.5–2022.11.26 | 2022.7.5–2022.11.28 | 2022.5.3–2022.11.22 | 2022.5.9–2022.11.24 | 2022.5.3–2022.12.24 |

The table shows the time ranges of Landsat 8 imagery acquisition in five regions of Tibet across three years (2016, 2019, and 2022). Data were retrieved from Google Earth Engine (GEE) and selected to ensure seasonal consistency across regions.

**Table 2. Model runtime environment and hyperparameter settings.**

| Region | Hyperparameter | Ranges | Runtime environment |
|---|---|---|---|
| **XGB** | Max Depth | 3–7 | Windows 11 x64 |
| | Min Child Weight | 1–2 | CPU: Intel Core i7–13700KF |
| | Gamma | 0.01–0.3 | RAM: 24G × 2 (DDR5, 6400 MT/s) |
| | Subsample | 0.6–0.95 | GPU: NVIDIA GeForce RTX 3080 (12G) |
| | Colsample by Tree | 0.35–1 | Python 3.8 |
| | Eta | 0.17–0.26 | MATLAB R2022b |
| | Lambda | 0–0.4 | |
| **RFR** | Min Samples Leaf | 10–25 | |
| | Number of Trees | 1–10 | |
| | Bootstrap | 0.1–1 | |
| **PLSR** | Stops adding independent-variable ($x$) when SSE < 0.0975 | Fixed | |

This table summarizes the hyperparameter ranges and runtime environment for the XGB, RFR, and PLSR models. All models were executed on a consistent hardware and software platform to ensure reproducibility and comparability.

## Formulas and explanations

**Evaluation metrics for model accuracy.** In evaluating model accuracy, the Coefficient of Determination ($R^2$) is a critical metric for assessing model quality. The Root Mean Square Error (RMSE) and Mean Absolute Error (MAE) are essential indicators for gauging model error. Consequently, a model demonstrating superiority should manifest high $R^2$ values concurrently with low RMSE and MAE values.

$$R^2 = 1 - \sum_{i=1}^{n} \frac{(y_i - \hat{y}_i)^2}{(y_i - \bar{y})^2} \tag{1}$$

$$RMSE = \sqrt{\frac{1}{n} \sum_{i=1}^{n} (y_i - \hat{y}_i)^2} \tag{2}$$

$$MAE = \frac{1}{n} \sum_{i=1}^{n} |\hat{y}_i - y_i| \tag{3}$$

In the equations above: $y_i$ represents the actual measured values, $\hat{y}_i$ denotes the predicted values, $\bar{y}$ is the mean of all measured values, $n$ is the number of samples. Additionally, STDEV refers to the standard deviation of the actual values in the testing dataset.

**Spectral indexes and environmental auxiliary variables.** Forest carbon density is generally believed to be closely related to vegetation growth, and soil quality degradation will directly affect vegetation growth, water retention capacity, and the content of various components in the soil (such as organic matter, soil carbon, and clay). Therefore, this study uses spectral information, indices, and environmental auxiliary variables in Landsat images as a feature candidate set and improves forest carbon density estimation accuracy through feature selection.

Specifically, the spectral indexes employed include the Soil Adjusted Vegetation Index (SAVI), its modified version (Modified Soil Adjusted Vegetation Index (MSAVI)), and the Normalized Difference Soil Organic Carbon (NDSOC) Index [43]. Their formulas are as follows:

$$SAVI = \frac{NIR - R}{NIR + R + L} \times (1 + L) \tag{4}$$

$$MSAVI = \frac{-\sqrt{(2NIR + 1)^2 - 8(NIR - R)} + 2NIR + 1}{2} \tag{5}$$

$$NDSOC = \frac{NIR - SWIR_a}{NIR + SWIR_a} \tag{6}$$

In the above formulas, $R$ represents the red band, $NIR$ represents the near-infrared band, $SWIR_a$ represents Band 6 of the short-wave infrared, and $L$ is set to 0.5. The calculation of the NDSOC index is determined based on the maximum correlation coefficient between soil organic matter (SOM) and each band.

The environmental auxiliary variables encompass the Normalized Difference Vegetation Index (NDVI), which is associated with vegetation health [44], along with the Enhanced Vegetation Index (EVI) [45]. For indexes indicating surface characteristics like Land Surface Brightness (LSB), Land Surface Greenness (LSG), and Land Surface Wetness (LSW) [20,46, 47], the formulas are as follows:

$$NDVI = \frac{NIR - R}{NIR + R} \tag{7}$$

$$EVI = 2.5 \times \frac{NIR - R}{1 + NIR + 6R - 7.5B} \tag{8}$$

$$LSB = Band_{mean} \tag{9}$$

$$LSG = NIR - G \tag{10}$$

$$LSW = NIR - SWIR_a \tag{11}$$

In the calculation of LSB, to fully reflect the surface brightness, we included all non-thermal infrared bands (B1–B7). We calculated LSG using a simplified vegetation indicator, which shows the difference between the near-infrared and green light bands. This difference calculation can effectively reveal the presence of vegetation because the reflectance of vegetation in the near-infrared band is generally higher and lower in the green light band. For the humidity index (LSW), we used the difference between the near-infrared and short-wave infrared bands as an indicator because these two bands are more sensitive to moisture, reflecting the surface humidity conditions. Therefore, $NIR$ is the near-infrared band, $B$ is the blue band, $R$ is the red band, $Band_{mean}$ represents the average reflectance of the B1–B7 bands, $G$ is the green band, and $SWIR_a$ is Band 6 of the short-wave infrared.

Regarding soil properties, the Soil Brightness Index (BI) and the Normalized Difference Moisture Index (NDMI) are selected to evaluate soil erosion and moisture conditions [48,49]; the formulas are as follows:

$$BI = \sqrt{R^2 + NIR^2} \tag{12}$$

$$NDMI = \frac{NIR - SWIR}{NIR + SWIR} \tag{13}$$

Where $NIR$ is the near-infrared band, $R$ is the red band, and $SWIR$ is the average reflectance of the B6 and B7 short-wave infrared bands.

Additionally, to monitor soil salinization, the Soil Salinity Index (SI) and the Soil Salinity Remote Sensing Monitoring Index (SDI) are utilized, effectively reflecting soil salinity [43,50, 51]; the formulas are as follows:

$$SI = \sqrt{B \times R} \tag{14}$$

$$SDI = \sqrt{\left(\frac{NIR - R}{NIR + R} - 1\right)^2 + SI^2} \tag{15}$$

Where $B$ is the blue band, $R$ is the red band, and $NIR$ is the near-infrared band.

Finally, for assessing soil parameters, the Clay Index (CI) is chosen [52,53]. These indexes, together with the original spectral data, form a comprehensive feature set for modelling; the formula is as follows:

$$CI = \frac{SWIR_a}{SWIR_b} \tag{16}$$

Where $SWIR_a$ and $SWIR_b$ represent B6 and B7 of the short-wave infrared, respectively.

**Explanations for normalization of metrics.** After standardization, the maximum value in the raw data is mapped to value $b$, while the minimum value is mapped to value $a$. The standardization formula is as follows:

$$x_{norm} = a + \frac{(x - x_{min})(b - a)}{x_{max} - x_{min}} \tag{17}$$

Where $x_{norm}$ represents the normalized value, $x$ is the original data value, and $x_{min}$ and $x_{max}$ are the minimum and maximum values in the dataset, respectively. This formula ensures that the data are redistributed within the interval $[a,b]$, facilitating more uniform data processing for subsequent analytical procedures. This means that the influence of different scales is effectively eliminated, enhancing the rationality and efficacy of data comparisons.

$$x'_{norm} = b - \frac{(x - x_{min})(b - a)}{x_{max} - x_{min}} \tag{18}$$

In this formula, the maximum value of the dataset is mapped to $a$, and the minimum value is mapped to $b$. Here, $x'_{norm}$ denotes the normalized value, $x$ represents the original data value, $x_{max}$ is the maximum value in the dataset, and $x_{min}$ is the minimum value. In this study, $a$ is set to 0.2, and $b$ is set to 1.

The calculation processes in this study were carried out in MATLAB R2022b and Python 3.9 open-source libraries.

## Methods of feature selection

Given the high variability and potential nonlinear characteristics of forest carbon density distribution, we chose nonlinear Spearman correlation analysis [54–56] (accessed on 13 Aug 2024) instead of Pearson correlation analysis. For feature selection, we will comprehensively consider the correlation of each feature in the five regions, sort the features in descending order, and finally select the first few features with the highest correlation.

The experiment used three models to establish carbon density estimation models and subsequent evaluation and analysis to prevent the randomness of experimental results, using 5-fold cross-validation. Divide all datasets into five parts, taking one part as the test set and the other as the training set without repetition. Then, take the average of the metric values from the validation sets of the model as the final evaluation metric. Model performance evaluation adopts three metrics: $R^2$, RMSE, and MAE. Evaluate the estimation performance of the model based on larger $R^2$, as well as smaller RMSE and MAE.

All algorithms and models in this study were implemented using Python 3.8 and MATLAB R2022b.

# Results

## Sample statistical properties

We conducted a statistical analysis of the forest carbon density of samples in each region Fig 2. The analysis results showed that the coefficient of variation of samples in each region was more significant than 1, showing spatial heterogeneity and the data distribution deviated significantly from the normal distribution.

## Results of feature selection

The correlation analysis results in Fig 3 show differences in the correlation between spectral index, band information, environmental auxiliary variables, and forest carbon density in each region. Given the previous method description, we comprehensively considered the feature correlations of the five regions, sorted the features in descending order, and finally selected the top six most relevant features. These features include Band3, Band4, Band7, NDSOC, LSB, NDMI, and CI. In addition, we are incorporating geographic information (Geo) as an additional standard feature.

Regarding the production of the spatiotemporal distribution map of forest carbon density content, we used ArcGIS Pro software to classify the prepared Landsat 8 data, extract the pixel information of vegetation in the image, and finally use the optimal model for each region to estimate the forest carbon density content in each region.

## Modelling results

Table 3 displays the metric values of each model under various methods. Fig 4 compares the accuracy of each model on the test set (Formulas 17 and 18).

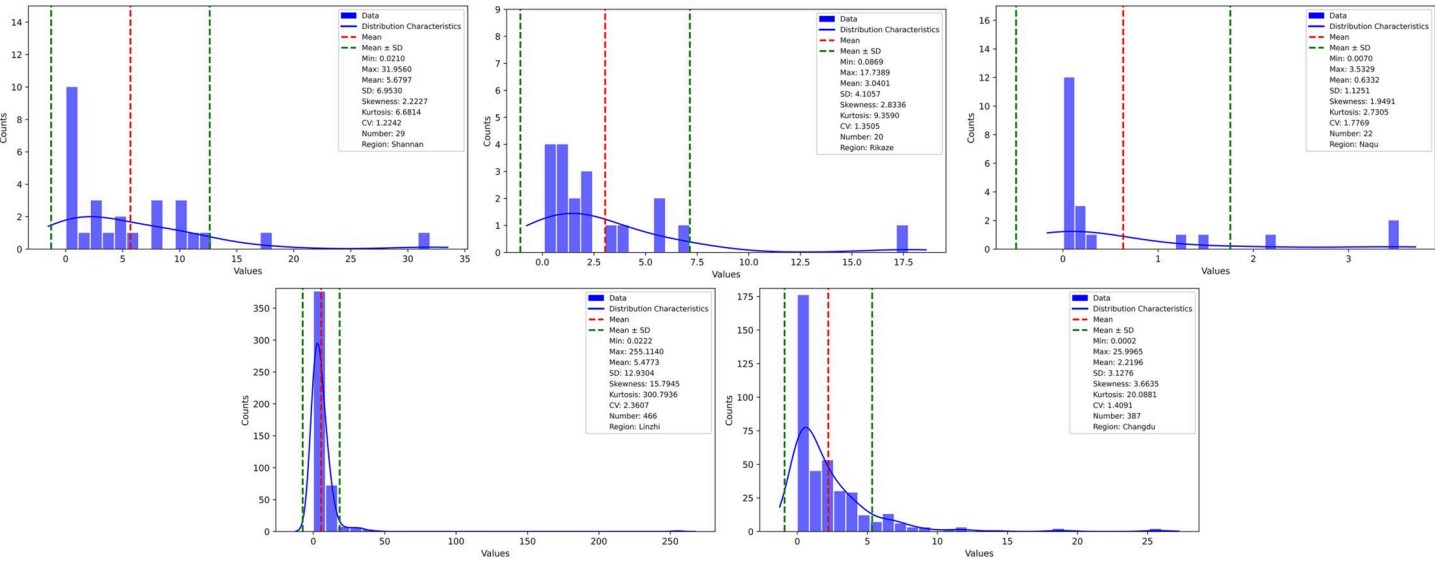

**Fig 2. Forest carbon density content information of samples (unit:t/ha).** This figure's horizontal axis represents the Forest carbon density content, and the vertical axis represents the number of samples. The histogram displays the number of samples within different Forest carbon density content ranges. The red dashed line marks the average Forest carbon density content in the samples, the green dashed line represents the range of SD near the mean value, and the blue line describes the distribution characteristics of Forest carbon density content in the samples. Additionally, the legend includes further statistical information about the Forest carbon density content in the samples.

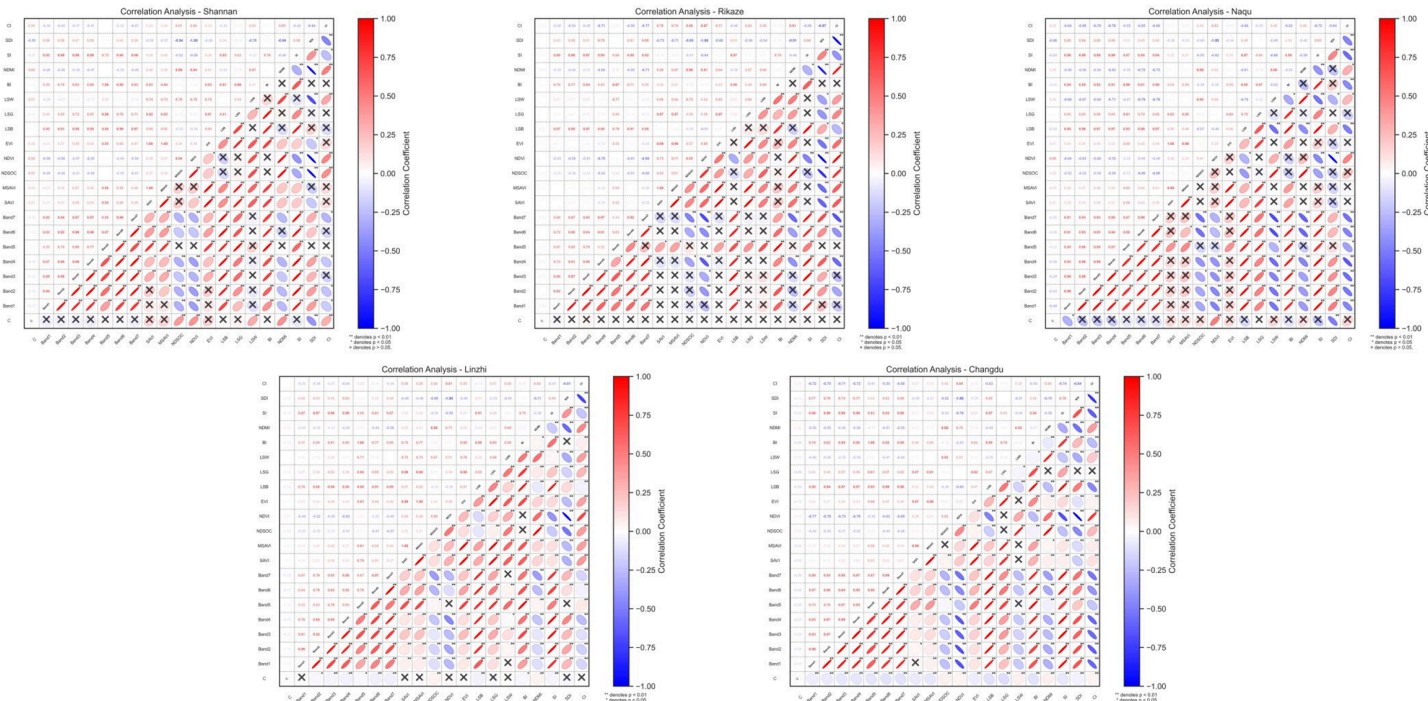

**Fig 3. The correlations of all features in each region.** Subfigures (a–e) show the correlation matrices for five regions: (a) Shannan, (b) Rikaze, (c) Naqu, (d) Linzhi, and (e) Changdu. The direction of the ellipse indicates the sign of the relationship, and its width and opacity reflect the mag-nitude of the relationship. Significance levels are denoted as ** for $p<0.01$, * for $p<0.05$, and × for $p>0.05$.

**Table 3. Modelling results.**

| Region | Model | Test Dataset | | |
|---|---|---|---|---|
| | | $R^2$ | RMSE (mg/kg) | MAE (mg/kg) |
| Changdu | XGB | 0.721 | 1.385 | 0.920 |
| | RFR | 0.598 | 1.526 | 1.184 |
| | PLSR | 0.324 | 1.918 | 1.592 |
| Linzhi | XGB | 0.684 | 1.760 | 1.179 |
| | RFR | 0.600 | 5.191 | 3.547 |
| | PLSR | 0.291 | 6.316 | 4.445 |
| Naqu | XGB | 0.702 | 0.407 | 0.360 |
| | RFR | 0.777 | 0.379 | 0.340 |
| | PLSR | 0.416 | 1.046 | 0.797 |
| Rikaze | XGB | 0.965 | 0.353 | 0.279 |
| | RFR | 0.674 | 0.485 | 0.331 |
| | PLSR | 0.455 | 1.026 | 1.072 |
| Shannan | XGB | 0.819 | 1.766 | 1.142 |
| | RFR | 0.630 | 2.202 | 1.963 |
| | PLSR | 0.562 | 3.011 | 2.500 |

The following conclusions can be drawn from Table 3 and Fig 4: (a) The nonlinear model occupies a dominant position in the prediction accuracy of each region, among which the performance of the XGB model is generally better than that of RFR and PLSR. However, in the Naqu region, RFR outperforms XGB, which may be related to the region-specific data

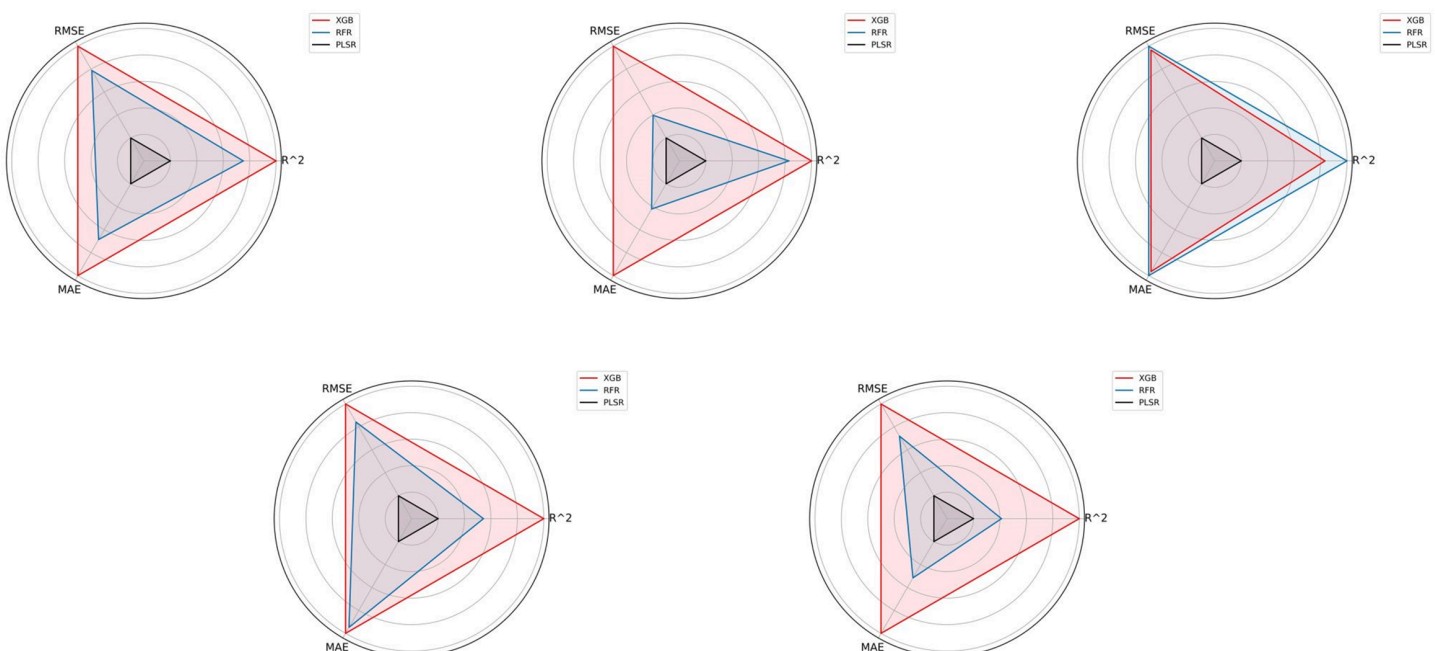

**Fig 4. Model accuracy comparison based on ground-based hyperspectral data (a–e).** Subfigures illustrate: (a) Changdu, (b) Linzhi, (c) Naqu, (d) Rikaze, and (e) Shannan. In this figure, $R^2$ is positively normalized (Eq 17), while RMSE and MAE undergo inverse normalization (Eq 18). The distance from the center reflects model accuracy for each metric, facilitating comparison across models and regions.

characteristics. (b) Due to differences in samples from different regions, the model's prediction errors in Naqu and Rikaze are relatively small, which shows that the data characteristics in these regions are more adaptable to the model, making the prediction results more accurate. (c) The model accuracy in the Rikaze and Shannan regions has the best performance. The $R^2$ value of the Rikaze region is as high as 0.96, which is significantly better than other regions, indicating that the model fitting effect in this region is particularly excellent.

Specifically, nonlinear models such as XGB and RFR have outstanding estimation performance and can effectively capture the complex nonlinear relationship between car-bon density and spectral reflectance. In comparison, most linear models have limited esti-mation accuracy, which can be explained by the advantages of nonlinear models in handling complex relationships. In-depth analysis shows that incorporating GEF and multispectral data as environmental auxiliary variables into feature inputs can significantly improve the accuracy of forest carbon density estimation. This shows that by fusing these environmental variables and geographical information, the model can effectively explain the complex nonlinear relationship between GEF and forest carbon density, thereby achieving high ac-curacy in estimating forest carbon density.

## Response and importance of each feature

Fig 5 shows the importance of different features in the optimal model for carbon density estimation in each region. Among them, Geo is identified as the most indispensable feature, always occupying the top two positions, especially in the Naqu region, where the importance of Geo accounts for more than 50%—followed by CI, ranking among the top three in all regions except Linzhi. The pie chart in the figure details the relative importance of GEF

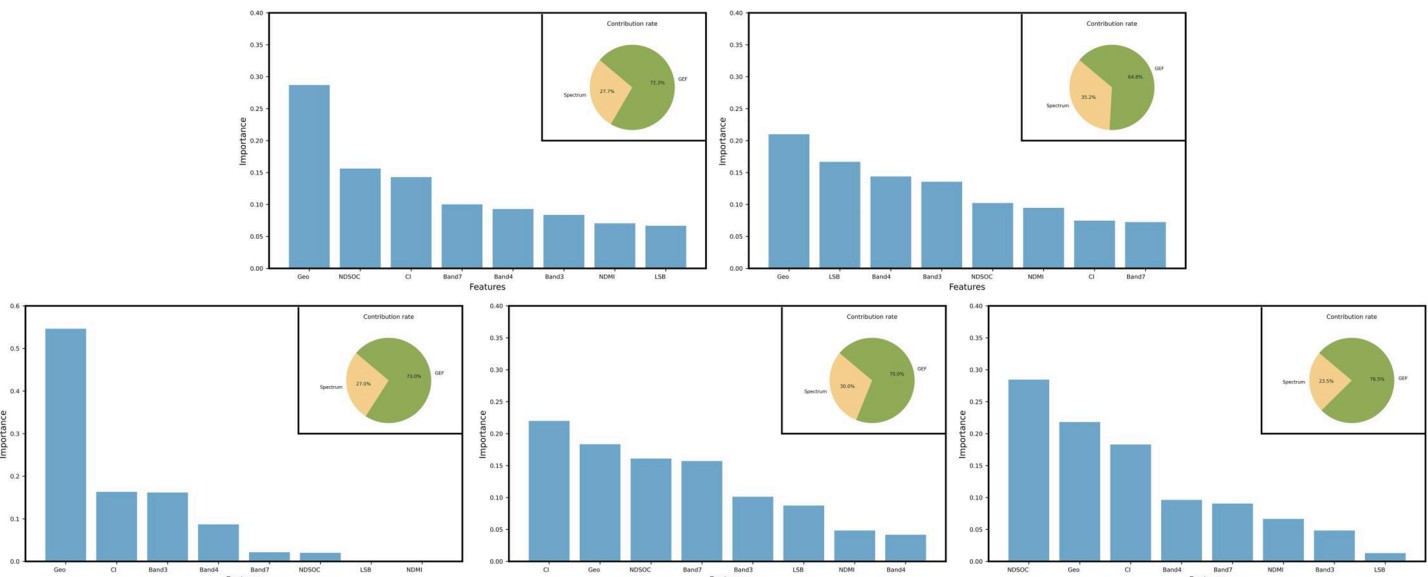

**Fig 5. The relative importance of each feature of the optimal model for each region (a-e).** Including (a) the XGB model of Changdu, (b) the XGB model of Linzhi, (c) the RFR model of Naqu, (d) the XGB model of Rikaze, and (e) XGB model of Shannan. In the figure presented, the column depicts the relative importance of individual features. At the same time, the pie chart illustrates the contribution rate (proportions of the relative importance) across various categories of features.

and Spectrum. Among them, GEF accounts for more than 60% of the overall importance, especially in the Shannan region, where the proportion of GEF exceeds 75%, highlighting the synergistic relationship between GEF and forest carbon density. This result shows that Geo features have significant effectiveness in carbon density estimation and are an indispensable key factor. At the same time, the relative importance of CI exceeds 10% on average across regions, which may be related to the direct impact of vegetation growth on forest carbon density. CI represents the clay content in the soil, and clay contributes to the absorption and assimilation of plant nutrients, resulting in a clear correlation between carbon density and CI, thereby improving the model's learning ability. These findings further validate the critical role of environmental auxiliary variables constructed from Geo and Landsat data in carbon density estimation and highlight their critical impact and effectiveness on forest carbon density prediction.

## Dynamic spatial distribution of forest carbon density from 2016 to 2022

According to Fig 6, the forest coverage rate in various regions of Xizang is generally high, except in the Naqu region. Between 2016 and 2022, the carbon density of each region changed slightly. It is worth noting that the Linzhi and Shannan regions have the highest carbon densities, located in their respective middle and lower right parts. In addition, Shannan and Linzhi are adjacent to each other. Specifically, the right side of Shannan is Linzhi. Therefore, the higher carbon density in these two regions can be explained. The Naqu region not only has the lowest forest coverage, but its carbon density is also significantly lower usually less than 1.5 t/ha. These analysis results highlight critical areas in Xizang that require forest management and can provide valuable reference information for relevant government departments.

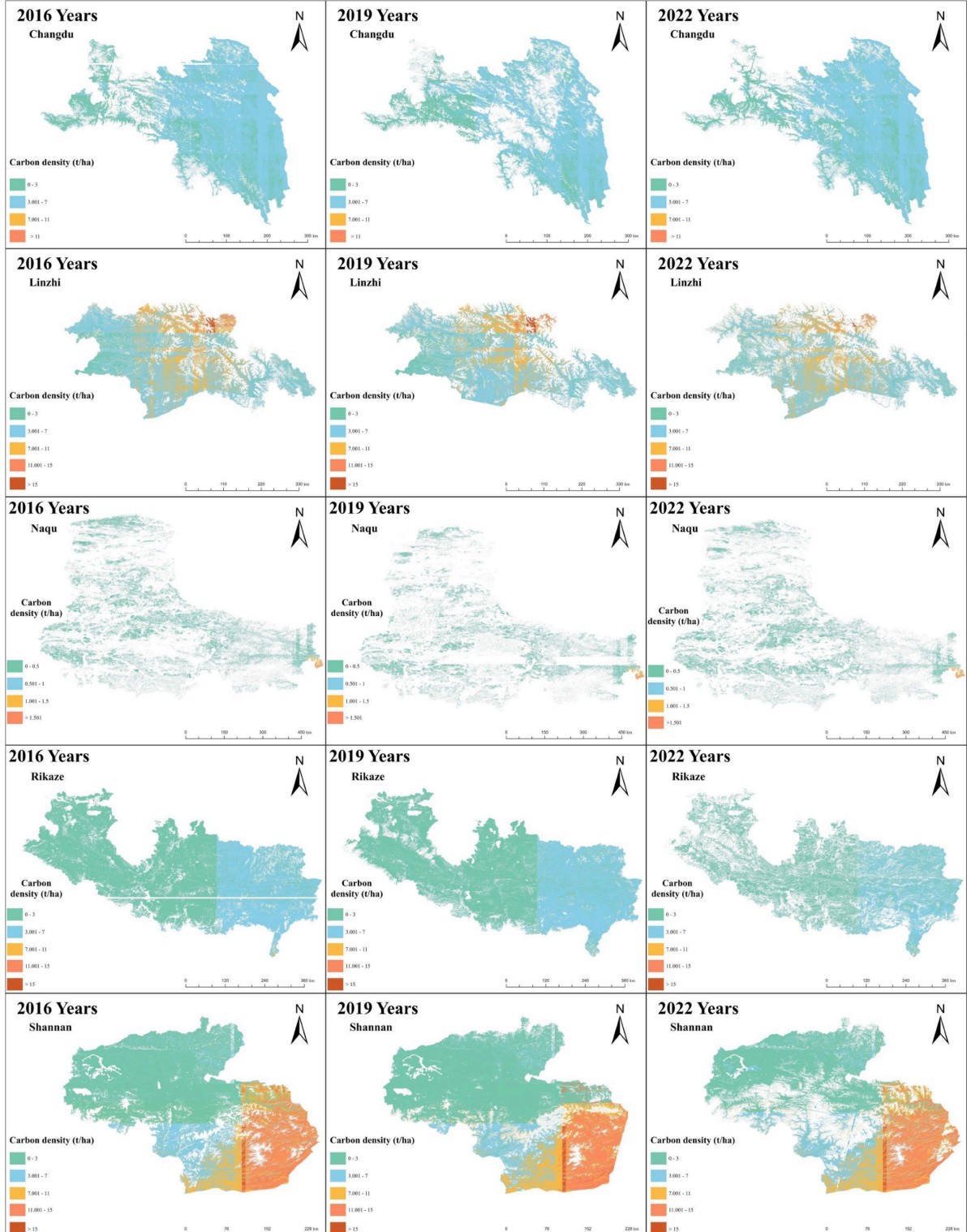

**Fig 6. Spatial and temporal distribution of Forest carbon density in Xizang, China from 2016 to 2022 (a-o).** Each submap shows the spatial distribution of forest carbon density for a specific year and region. Due to uncontrollable factors such as atmospheric conditions and cloud cover, the image coverage in some areas may slightly differ, such as in Changdu (2019), and Rikaze and Shannan (2022). Base map and provincial boundaries are from Natural Earth (public domain; https://www.naturalearthdata.com). Remote sensing imagery was acquired from Landsat OLI/TIRS data (public domain; https://landsat.visibleearth.nasa.gov/) for each corresponding year.

## Discussion

### Reasons for the response of the environmental factors to Forest carbon density

Forest carbon density is affected by a complex interaction of multiple environmental factors, which together determine the forest's carbon storage capacity. First, climatic factors such as temperature, precipitation, and humidity [57] directly affect plant growth. Suitable temperatures and sufficient precipitation often significantly promote photosynthesis of plants [58], thereby enhancing carbon fixation and storage capabilities. Temperature changes affect plant physiological activities [59], while changes in precipitation and humidity directly affect soil moisture supply [60]. The combined effect of these factors determines the carbon storage potential of forests. Second, soil properties also play an essential role in forest carbon density. The texture, fertility, and pH of the soil directly affect the growing environment of plants. Fertile soil can provide abundant nutrients to support healthy plant growth [61], thereby effectively increasing the carbon storage capacity of forests. At the same time, soil texture and pH significantly affect carbon accumulation and release [62], and the combined effect of these factors determines the overall carbon density of the forest. Finally, the impact of different vegetation types on carbon density is also apparent [63]. Biodiverse forests often exhibit higher carbon densities [64]. These forests possess dense canopies and enhance carbon sequestration and storage through complex ecosystem interactions [65]. Therefore, the diversity and coverage of vegetation types play a crucial role in carbon storage. In summary, the interaction between climatic conditions, soil characteristics, and vegetation types has a crucial impact on forest carbon density. This is why environmental auxiliary variables such as CI, NDMI, and NDSOC play a crucial role in this study. Therefore, understanding these related environmental factors will help promote the study of global carbon density with multispectral images and play an essential role in the balance of ecological cycles.

### Limitations and caveats

Although this study makes significant progress in forest carbon density monitoring, some limitations still deserve further discussion. First, the general applicability of the model has not been thoroughly verified. The data used in the current study mainly come from Xizang, China, which makes us have some doubts about the effectiveness of this method in other geographical areas. Therefore, future research should focus on testing and validating the adaptability and robustness of this method in different geographical landscapes to ensure its widespread application on a global scale. Second, while incorporating GEF data does provide a valid estimate of carbon density, the availability and quality of this data can vary significantly across regions. This data inhomogeneity may limit the method's effectiveness in regions where spectral data are scarce. Future work needs to consider how to optimize this approach in areas where data are scarce or explore how to supplement and enhance data sources to improve the accuracy and reliability of estimates. Finally, although the model has integrated multiple variables in its estimates, it has not taken into account some unmeasured factors that may affect the results, such as microclimate conditions and specific land use practices. These factors may have a significant impact on estimates of forest carbon density. Future research should consider incorporating these unmeasured factors into models to improve the accuracy and comprehensiveness of estimates further. In summary, although this study provides a powerful tool, its applicability, data dependence, and potential missing factors need to be overcome and optimized in future research.

### Implications for future research and applications

This study successfully developed a novel and highly versatile forest carbon density estimation method that combines GEF data with advanced machine learning algorithms. Through the application of this method, researchers and practitioners can estimate forest carbon density with optimal accuracy, providing crucial technical support for forest management and environmental protection. In addition, the successful implementation of this method has opened up new avenues for research and practical applications, especially in addressing human sustainability challenges. Specifically, similar techniques can be extended to other critical areas, such as soil carbon estimation, to further promote a deeper understanding and protection of the environment. The open-source tool developed based on this method will facilitate researchers and practitioners around the world to apply this method more conveniently for related research. The promotion and application of this tool will improve the research level in environmental protection and sustainability and strengthen international cooperation to jointly address global environmental challenges. Through this innovative technology and tool, we are expected to achieve more significant results in promoting scientific research and practical applications.

## Conclusions

Our findings from this study can be summarized as follows:

**Effectiveness of nonlinear techniques:** This study covers an area of 836,640 square kilometers. Based on multispectral image data, two nonlinear models (XGB and RFR) and one linear model (PLSR) were used to estimate forest carbon density by comprehensively considering geographic information, environmental variables, and spectral information. The results show that the XGB model performs significantly better than other models, although it is slightly less effective in Naqu. Moreover, especially in Rikaze, the $R^2$ of the model exceeds 0.96, while the comprehensive $R^2$ across regions exceeds 0.77. This result, combined with the vast study area, proves the importance of nonlinear models and shows that forest carbon density has a highly nonlinear relationship with geographic information, environmental variables, and spectral information, confirming the effectiveness of this study's experimental design.

**Effectiveness and importance of GEF:** Geo is the most critical factor, consistently ranking in the top two, especially in the Naqu area, where its importance exceeds 50%. Second, CI ranks in the top three regions except Linzhi. Overall, the relative importance of GEF exceeds 60%, and in the Shannan region, it exceeds 75%, emphasizing the close relationship between GEF and forest carbon density. The relative importance of CIs averages over 10% across regions. These results verify the critical role of environmental variables constructed from Geo and Landsat data in carbon density estimates, highlighting their significant impact on prediction models.

**Changes in forest carbon density in Xizang:** Most areas in Xizang have high forest coverage, except Naqu. Between 2016 and 2022, the carbon density of each region did not change much. Linzhi and Shannan had the highest carbon density, while Naqu had the lowest forest coverage and carbon density below 1.5 t/ha. This highlights the critical areas for forest management in Xizang and provides a reference for relevant government departments to formulate policies.

Our study successfully estimated the spatial distribution of carbon density over an extensive range of years based on multispectral data and proposed an effective carbon density estimation method (incorporating GEF information). At the same time, we determined the optimal model (XGB) and verified that the combination of the two has the best effect. This method effectively expands the application scope of multispectral data. It strongly supports

national policy formulation related to carbon sinks and critical technical support for the global forest carbon sink monitoring challenge.

## Acknowledgments

Our heartfelt thanks go to everyone who contributed to this project.

## Author contributions

**Conceptualization:** Jiang ping Fang.

**Data curation:** Li Cheng, Yang yang Xia, Wen wen Guo, Rui qiang Chi.

**Formal analysis:** Li Cheng, zi ling Yang.

**Methodology:** Li Cheng, Jiang ping Fang.

**Resources:** Jiang ping Fang.

**Software:** Li Cheng.

**Supervision:** Jiang ping Fang.

**Visualization:** Li Cheng.

**Writing – original draft:** Li Cheng.

**Writing – review & editing:** Jiang ping Fang.

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
