## [Decision Letter · Decision Letter 0]

8 Apr 2025

PONE-D-25-14912Mapping spatiotemporal distribution of forest carbon density in Tibet, ChinaPLOS ONE

Dear Dr. Cheng,

Thank you for submitting your manuscript to PLOS ONE. After careful consideration, we feel that it has merit but does not fully meet PLOS ONE’s publication criteria as it currently stands. Therefore, we invite you to submit a revised version of the manuscript that addresses the points raised during the review process. Please submit your revised manuscript by May 23 2025 11:59PM. If you will need more time than this to complete your revisions, please reply to this message or contact the journal office at plosone@plos.org. Please include the following items when submitting your revised manuscript:

We look forward to receiving your revised manuscript.

Kind regards,

Upaka Rathnayake, PhD

Academic Editor

PLOS ONE

“Authors�LC,ZLY, YYX,WWG,RQC,JPF

Project title and grant number:Spatial Patterns of Carbon Sinks in Tibetan Forest Ecosystems and the Factors Affecting Them(XZ202401ZY0090);Tibet University Graduate High-level Talent Cultivation Research Fund Project(2022-GSP-B011)

Funding unit Tibet Science and Technology Agency Tibet University”

5. For studies involving third-party data, we encourage authors to share any data specific to their analyses that they can legally distribute. PLOS recognizes, however, that authors may be using third-party data they do not have the rights to share. When third-party data cannot be publicly shared, authors must provide all information necessary for interested researchers to apply to gain access to the data. (https://journals.plos.org/plosone/s/data-availability#loc-acceptable-data-access-restrictions)

4) All necessary contact information others would need to apply to gain access to the data.

6. PLOS requires an ORCID iD for the corresponding author in Editorial Manager on papers submitted after December 6th, 2016. Please ensure that you have an ORCID iD and that it is validated in Editorial Manager. To do this, go to ‘Update my Information’ (in the upper left-hand corner of the main menu), and click on the Fetch/Validate link next to the ORCID field. This will take you to the ORCID site and allow you to create a new iD or authenticate a pre-existing iD in Editorial Manager.

7. We note that Figures 1 and 6 in your submission contain [map/satellite] images which may be copyrighted. All PLOS content is published under the Creative Commons Attribution License (CC BY 4.0), which means that the manuscript, images, and Supporting Information files will be freely available online, and any third party is permitted to access, download, copy, distribute, and use these materials in any way, even commercially, with proper attribution. For these reasons, we cannot publish previously copyrighted maps or satellite images created using proprietary data, such as Google software (Google Maps, Street View, and Earth). For more information, see our copyright guidelines: http://journals.plos.org/plosone/s/licenses-and-copyright.

1. You may seek permission from the original copyright holder of Figures 1 and 6 to publish the content specifically under the CC BY 4.0 license. 

Reviewers' comments:

Reviewer's Responses to Questions

**Comments to the Author**

1. Is the manuscript technically sound, and do the data support the conclusions?

Reviewer #1: Yes

Reviewer #2: Yes

2. Has the statistical analysis been performed appropriately and rigorously? 

Reviewer #1: Yes

Reviewer #2: Yes

3. Have the authors made all data underlying the findings in their manuscript fully available?

Reviewer #1: No

Reviewer #2: Yes

4. Is the manuscript presented in an intelligible fashion and written in standard English?

Reviewer #1: Yes

Reviewer #2: Yes

5. Review Comments to the Author

Reviewer #1: The manuscript titled "Mapping Spatiotemporal Distribution of Forest Carbon Density in Tibet, China" engages with a critical environmental subject using advanced analytical methods. The authors apply Landsat 8 data alongside machine learning techniques to evaluate forest carbon density, offering a valuable contribution to the fields of remote sensing and environmental science. Although the manuscript is generally well-organized and presented, there are several areas where improvements could enhance clarity and the robustness of the study's conclusions.

Specific Comments

Abstract

The abstract is well-organized and structured, effectively summarizing the findings of the study.

Introduction

The introduction is well written but could be enhanced by discussing more recent studies that have used similar methodologies or focused on similar regions to enhance the contextual backdrop of the research.

Materials and Methods

The selection and description of the study area are detailed; however, the manuscript would benefit from a more comprehensive explanation of the criteria used for sample selection and the rationale behind choosing specific regions within Tibet.

Additionally, the explanation of the data processing steps is adequate, but the authors should consider providing a more detailed description of the machine learning models used, including any parameter tuning or validation processes.

Discussion

The manuscript benefits from a thoughtful discussion, particularly regarding the implications of the findings. However, it could be enhanced by including a comparison with other global regions where similar studies have been conducted to broaden the relevance of the findings.

Conclusion

The conclusions are well-summarized; however, they could be strengthened by explicitly stating how this research advances our understanding of forest carbon dynamics and suggesting specific areas for further research or policy implementation.

Grammar and Spelling

Overall, the manuscript is well-written. However, thorough proofreading needs to be done to correct minor typographical errors and ensure consistency in terminology. Specifically, attention should be paid to the hyphenation and spelling of terms like "multi-spectral" which appears inconsistently throughout the text.

Reviewer #2: Review of Mapping spatiotemporal distribution of forest carbon density in Tibet, China

Overall Recommendation: Major Revision

The manuscript presents a valuable contribution to regional carbon density estimation using machine learning and geospatial indices. The methodology is sound, the use of open-source tools is commendable, and the findings offer clear ecological and policy relevance. However, some major/minor but important revisions are necessary to enhance clarity, methodological justification and interpretation of results.

These include:

- Strengthening the justification for feature selection,

- Deepening the explanation of regional model performance,

- Expanding the discussion of limitations and future improvements.

Once these revisions are addressed, the manuscript will be suitable for publication.

Suggestions for Introduction Section

Line 45: Solid opening statement, but consider being more specific about what aspects of climate change (e.g., temperature rise, carbon imbalance) are being addressed.

Line 46: Good point, but "irreplaceable" could be rephrased to avoid exaggeration (e.g., "play a vital role").

Line 47: Slightly redundant — since it's known that trees are the main part of forests.

Line 48–49: Strong statement on services and carbon, but "thereby significantly affecting..." is a bit repetitive with earlier lines. Consider simplifying.

Line 50: "Will decrease" should be "decreased" (tense mismatch, you're referring to 2015–2020).

Line 51–52: Well positioned, though "self-evident" is subjective. Try rewording to “widely recognized.”

Line 53–54: Sentence is meaningful but wordy. Consider breaking into two for impact.

More clarity is needed on how carbon estimation differs between methods. Add examples or references for better clarity (e.g., which geostatistical model? which machine learning algorithm?). Define all acronyms clearly and early (e.g., GEF, SAR).

Missing Elements

i. A clear research gap and novelty statement.

ii. Rationale for using Landsat 8 over other sensors like Sentinel-2.

iii. Short mention of expected accuracy or comparison with traditional methods.

iv. Clarify how machine learning is applied (brief method, like RF, SVM?).

Suggestion to study area Section

The section lacks ecological context; Tibet’s role in carbon dynamics should be clearly stated. Climatic variation is described vaguely; use specific regions and quantify conditions. Long, complex sentences reduce clarity and need restructuring. Population and forest data are disconnected from the study focus explain their relevance to carbon estimation. Statistical claims lack citations, weakening credibility. The paragraph is descriptive but not analytical; link geographic and climatic features directly to the study’s objectives.

Suggestion to Sample acquisition and statistical properties

Line 117: "From May to the end of June 2016..."

Please Specify why this time frame was chosen. Is it optimal for biomass measurement (e.g., growing season)? Also, mention whether the sampling was one-time or part of a long-term dataset.

Line 118: "total sampling area of 836,640 square kilometers..."

That’s an extremely large area (nearly all of Tibet). Clarify if this refers to total area covered by all regions or the spatial distribution extent. Sampling intensity (e.g., per sq km) is unclear.

Line 119: "recorded the GPS coordinate information..."

What was the accuracy level of GPS devices used? Were coordinates collected in WGS84 or a different projection?

Lines 120–122: Good detail on DBH measurement standard. However, mention the tree selection criteria (e.g., random, dominant species, spacing), and how many trees were sampled per plot.

Lines 123–125: "...accurate to 0.1 cm... converted it into diameter."

Please Include potential sources of error here. Were double measurements or QA/QC checks performed?

Lines 126–127: Cleaning before measurement is good practice. You could add if saplings or trees below a certain DBH threshold were excluded.

Line 128: "forest carbon density was calculated in the laboratory..."

This is vague. What method was used (e.g., allometric equations, biomass expansion factors)? Was species-specific carbon fraction used?

Line 129: "Due to the wide distribution..."

Need Justification for choosing these five administrative regions should be added. Were ecological zones or forest types also considered?

Suggestion to Method of feature selection

Line 223 "Given the high variability and potential nonlinear characteristics of forest carbon density distribution..."

Suggestion: Briefly explain why forest carbon density is nonlinear (e.g., influenced by complex ecological interactions and environmental heterogeneity). Support with a scholarly reference.

Line 224–225 "...we chose nonlinear Spearman correlation analysis..."

Issue: A citation from Wikipedia is used.

Suggestion: Replace the Wikipedia citation with a peer-reviewed journal or statistical textbook for academic credibility.

Line 226–229 "...we will comprehensively consider the correlation of each feature..." Issue: The term “features” is vague.

Suggestion: Clearly specify what features were considered (e.g., spectral bands, vegetation indices, topographic variables). Explain how many features were finally selected, and whether a threshold correlation coefficient was applied.Mention if multicollinearity between features was checked (important in machine learning).

Line 230 "The experiment used three models..."

Issue: Models are unnamed.

Suggestion: Specify which three models were used (e.g., Random Forest, XGBoost, SVR) and why they were chosen for comparison.

Line 231–233"...using 5-fold cross-validation. Divide all datasets into five parts..."

Cross-validation is appropriate.

Suggestion: Mention whether stratified cross-validation was used, especially if the data distribution is imbalanced. Clarify if a random seed was set for reproducibility.

Line 234–236 "...evaluate the estimation performance... larger R², smaller RMSE and MAE."

Good use of standard evaluation metrics.

Suggestion: Briefly define R², RMSE, and MAE to ensure clarity for all readers. State whether performance was averaged across folds or if model performance varied significantly across regions.

Line 237–238 "All algorithms and models... in Python 3.8 and MATLAB R2022b."

Mentioning software is good practice.

Suggestion: Add specific packages/libraries used (e.g., scikit-learn, NumPy, MATLAB’s Statistics Toolbox) for reproducibility.

3.1 Sample Statistical Properties

- High coefficient of variation (>1) confirms spatial heterogeneity.

- Non-normal distribution affects model assumptions.

- Lacks discussion of regional outliers or implications on modeling.

3.2 Feature Selection

- Smart integration of spectral (NDMI, Band3/4/7) and GEF features (CI, LSB, Geo).

- Geo chosen universally – reflects spatial dominance.

- Needs more statistical justification for feature choices.

3.3 Modelling Results

- XGB performs best overall, especially in Rikaze (R² = 0.965).

- RFR outperforms in Naqu – suggests local model–data fit matters.

- No mention of model complexity or training efficiency.

3.4 Feature Importance

- Geo is the dominant variable in most regions (>50% in Naqu).

- CI is consistently influential – links soil and carbon density.

- Good visuals, but regional differences need deeper exploration.

3.5 Dynamic Spatial Distribution

- Carbon density stable from 2016–2022; Linzhi and Shannan highest.

- Naqu remains lowest – clear ecological intervention target.

- Trend analysis is missing – no inter-annual change patterns.

4.1 Environmental Factor Response

- Explains role of climate and soil well – aligns with feature importance.

- Strong conceptual foundation.

- Could use more data-specific examples to connect theory to results.

4.2 Limitations and Caveats

- Acknowledges regional specificity and missing variables (e.g., microclimate).

- Transparency is strong.

- Should propose methods to integrate more diverse data.

4.3 Future Research and Applications

- Open-source tool offers practical value.

- Pushes for scaling and interdisciplinary work.

- Needs clarity on how tool will be shared, maintained, or validated globally.

5. Conclusion

- Models + GEF = strong prediction power.

- Geo and CI are core features.

- Solid wrap-up, but could briefly suggest satellite integration or future tech adoption.

6. PLOS authors have the option to publish the peer review history of their article (what does this mean?). If published, this will include your full peer review and any attached files.

Reviewer #1: No

Reviewer #2: No

---

## [Author Response · Author response to Decision Letter 1]

10 Jun 2025

Dear Editors and Reviewers,

Thank you for your thoughtful comments and suggestions. We have carefully revised the manuscript based on the feedback provided. The detailed responses to each reviewer comment have been submitted in a separate document, which addresses all the points raised.

We hope these revisions adequately address your concerns and look forward to your feedback.

---

## [Decision Letter · Decision Letter 1]

7 Sep 2025

Mapping spatiotemporal distribution of forest carbon density in Tibet, China

PONE-D-25-14912R1

Dear Dr.Cheng,

We’re pleased to inform you that your manuscript has been judged scientifically suitable for publication and will be formally accepted for publication once it meets all outstanding technical requirements.

Kind regards,

Laxmi Kant Sharma, PhD

Academic Editor

PLOS ONE

Additional Editor Comments (optional):

Reviewer #1:

Reviewers' comments:

Reviewer's Responses to Questions

**Comments to the Author**

1. If the authors have adequately addressed your comments raised in a previous round of review and you feel that this manuscript is now acceptable for publication, you may indicate that here to bypass the “Comments to the Author” section, enter your conflict of interest statement in the “Confidential to Editor” section, and submit your "Accept" recommendation.

Reviewer #1: All comments have been addressed

2. Is the manuscript technically sound, and do the data support the conclusions?

Reviewer #1: Yes

3. Has the statistical analysis been performed appropriately and rigorously? 

Reviewer #1: N/A

4. Have the authors made all data underlying the findings in their manuscript fully available?

Reviewer #1: Yes

5. Is the manuscript presented in an intelligible fashion and written in standard English?

Reviewer #1: Yes

6. Review Comments to the Author

Reviewer #1: (No Response)

7. PLOS authors have the option to publish the peer review history of their article (what does this mean?). If published, this will include your full peer review and any attached files.

Reviewer #1: No

---

## [Editor Report · Acceptance letter]

PONE-D-25-14912R1

PLOS ONE

Dear Dr. Cheng,

I'm pleased to inform you that your manuscript has been deemed suitable for publication in PLOS ONE. Congratulations! Your manuscript is now being handed over to our production team.

Kind regards,

on behalf of

Prof.Dr. Laxmi Kant Sharma

Academic Editor

PLOS ONE